# Natural genetic variation underlying the negative effect of elevated CO₂ on ionome composition in *Arabidopsis thaliana*

Oceane Cassan[1], Lea-Lou Pimpare[1], Timothy Mozzanino[1], Cecile Fizames[1], Sebastien Devidal[2], Fabrice Roux[3], Alexandru Milcu[2,4], Sophie Lebre[5], Alain Gojon[1], Antoine Martin[1]*

[1]IPSiM, Univ Montpellier, CNRS, INRAE, Institut Agro, Montpellier, France; [2]Montpellier European Ecotron, Univ Montpellier, CNRS, Campus Baillarguet, Montpellier, France; [3]Laboratoire des Interactions Plantes-Microbes-Environnement, Institut National de Recherche pour l'Agriculture, l'Alimentation et l'Environnement, CNRS, Université de Toulouse, Castanet-Tolosan, France; [4]CEFE, Univ Montpellier, CNRS, EPHE, IRD, Montpellier, France; [5]IMAG, Univ Montpellier, CNRS, Montpellier, France

*For correspondence:
antoine.martin@cnrs.fr

Competing interest: The authors declare that no competing interests exist.

**Abstract** The elevation of atmospheric CO₂ leads to a decline in plant mineral content, which might pose a significant threat to food security in coming decades. Although few genes have been identified for the negative effect of elevated CO₂ on plant mineral composition, several studies suggest the existence of genetic factors. Here, we performed a large-scale study to explore genetic diversity of plant ionome responses to elevated CO₂, using six hundred *Arabidopsis thaliana* accessions, representing geographical distributions ranging from worldwide to regional and local environments. We show that growth under elevated CO₂ leads to a global decrease of ionome content, whatever the geographic distribution of the population. We observed a high range of genetic diversity, ranging from the most negative effect to resilience or even to a benefit in response to elevated CO₂. Using genome-wide association mapping, we identified a large set of genes associated with this response, and we demonstrated that the function of one of these genes is involved in the negative effect of elevated CO₂ on plant mineral composition. This resource will contribute to understand the mechanisms underlying the effect of elevated CO₂ on plant mineral nutrition, and could help towards the development of crops adapted to a high-CO₂ world.

## eLife assessment

This paper provides **useful** information about how the ionome of *Arabidopsis thaliana* adapts to very high CO2-levels, backed up by **solid** evidence and carefully designed studies. The work will be of interest to anyone studying natural genetic variation as well as the response of plants to altered CO2 levels in the atmosphere.

## Introduction

The elevation of atmospheric CO₂ concentration leads to a decline in the mineral composition of C3 plants (*Gojon et al., 2023*). The negative effect of elevated CO₂ on plant mineral composition has been observed worldwide, and alters the content of nutrients that are essential for human

nutrition, such as nitrogen (N) and proteins, iron (Fe) or zinc (Zn) (*Loladze, 2014*). Therefore, the rise in atmospheric $CO_2$ poses a significant threat to food security in the coming decades. Indeed, several modeling approaches predict a decrease in plant-based nutrient availability due to the negative effect of elevated $CO_2$ on the mineral status of plants, leading to an additional risk of nutritional deficiency for hundreds of millions of people (*Medek et al., 2017*; *Ebi and Loladze, 2019*). However, the reasons why elevated $CO_2$ leads to the degradation of plant mineral composition are far from being well understood. To date, only a few genes with a potential regulatory role on this effect have been identified (*Gao et al., 2019*; *Umnajkitikorn et al., 2020*; *Yang et al., 2020*; *Bouain et al., 2022*; *Sun et al., 2022*; *Cassan et al., 2023*). Despite their limited number, these studies show that the detrimental effect of high $CO_2$ on the plant mineral status has genetic bases. In addition to this, several reports suggest that exploring the natural genetic variability of plants represents a major opportunity to understand the mechanisms by which high $CO_2$ leads to a decline in plant mineral composition (*Myers et al., 2014*; *Zhu et al., 2018*; *Marcos-Barbero et al., 2021*). Indeed, a significant diversity in the response of mineral composition to high $CO_2$ has been observed in several plant species. For protein and therefore N content, as well as for Fe or Zn content, substantial variations have been observed between small panels of genotypes from different species (*Myers et al., 2014*; *Zhu et al., 2018*; *Marcos-Barbero et al., 2021*). This implies the presence of a genetic diversity reservoir, which can facilitate the understanding of the ionome's response to high $CO_2$ and subsequently provide an opportunity to alleviate this negative impact. However, in order to identify the genetic determinants of this negative response of the ionome to high $CO_2$, large-scale approaches are necessary, but are still lacking for the moment. The objective of this work was to fill the aforementioned knowledge gap by using a large collection of natural genotypes of the model plant *Arabidopsis thaliana*. This allowed to explore in depth the natural variation of the ionome response to elevated $CO_2$, and to generate a resource of phenotypic data that can be used in association genetics approaches. To this end, we used several hundreds of accessions from different geographic scales of *A. thaliana*, and analyzed their leaf mineral composition under contrasted conditions of $CO_2$ concentration. This allowed us to extract the general trends in the leaf ionome response to high $CO_2$, and to identify a large set of genes associated with the variation in the mineral composition of plants in response to high $CO_2$. By combining this information with genome expression data under elevated $CO_2$, we end up by functionally validating one of these genes for its importance in the reduction of Zn content under elevated $CO_2$. Our results open the way for a better understanding of the genetic and molecular mechanisms involved in the regulation of plant mineral nutrition by the elevation of atmospheric $CO_2$.

## Results

In order to explore the natural variation and identify its underlying genetic basis associated with the negative effect of elevated $CO_2$ on plant ionome, we used three populations of *A. thaliana* representing different geographic scales (i.e. the worldwide REGMAP population, the LANGUEDOC regional population and the local TOU-A population from east of France) and displaying different levels of genetic diversity (*Figure 1A*). These populations were grown under ambient or elevated $CO_2$, and we measured in each accession the composition of their ionome in rosettes, including C, N, Na, Fe, Mg, Mn, Zn, and Cu content.

### Elevated $CO_2$ globally decreases ionome content at the population level, whatever the geographic scale

In the three *A. thaliana* populations, we observed a global and important decrease of the ionome content when plants were grown under elevated $CO_2$ as compared to ambient $CO_2$. This was particularly the case for N and Fe, for which the decrease in content was very robust and important in each of the population analyzed (*Figure 1B–D*). Zn, Cu, and Mg content were also negatively affected to a significant extent by the growth under elevated $CO_2$ in the REGMAP and in the TOU-A populations (*Figure 1B and C*), although not significantly in the LANGUEDOC population (*Figure 1D*). More variability for the effect of elevated $CO_2$ was observed on Mn and Na content, which were decreased in the REGMAP population, but not significantly changed in the TOU-A and LANGUEDOC populations, respectively. In parallel, the C content of these populations increased under elevated $CO_2$, by very significant factors for the REGMAP and the LANGUEDOC populations. Altogether, these observations

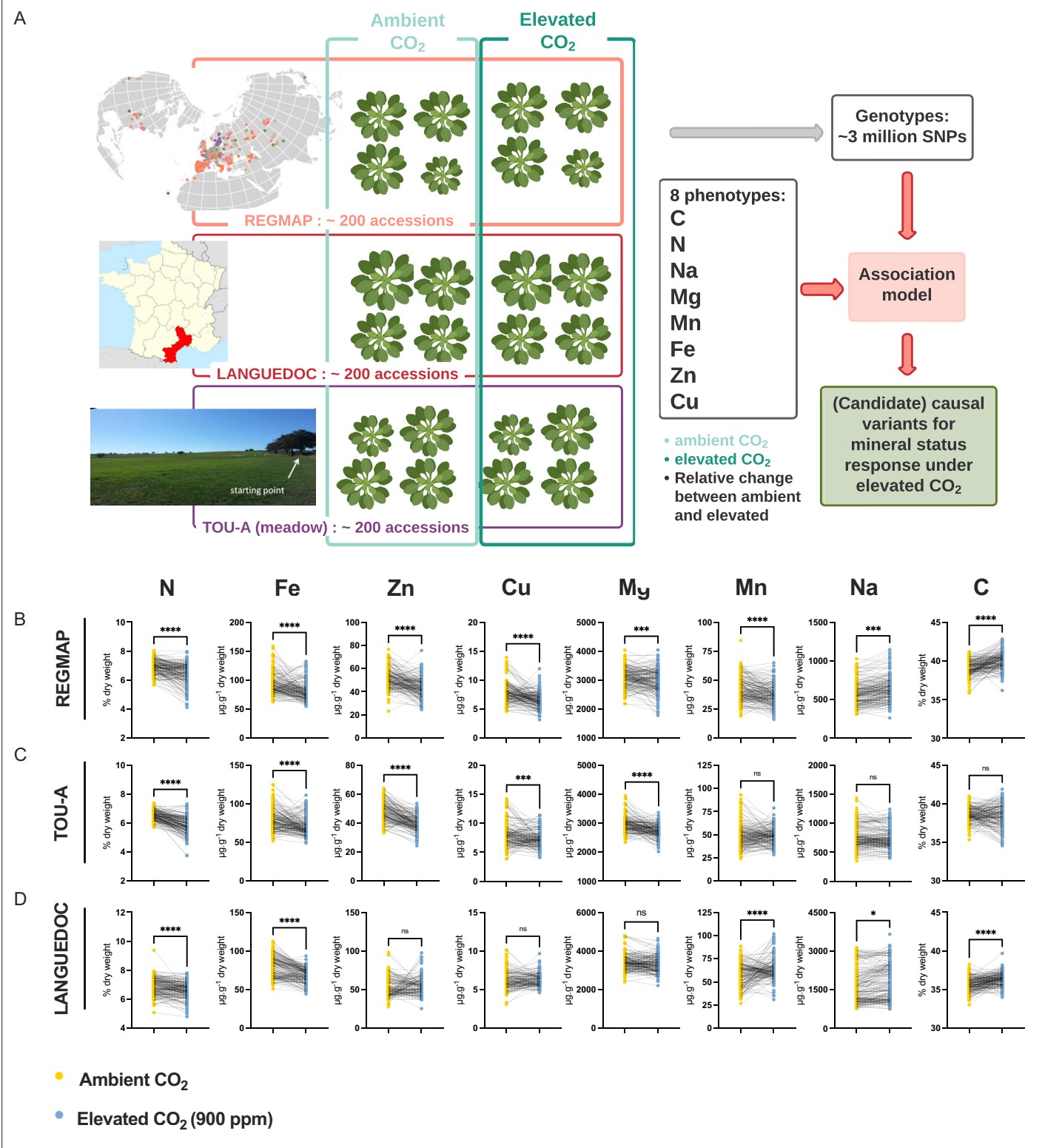

**Figure 1.** Elevated $CO_2$ negatively impacts the ionome content at the population-scale level in *Arabidopsis thaliana*. (**A**) Representation of the experimental design used in this study. The content of eight mineral elements was assessed for around 600 *Arabidopsis thaliana* accessions coming from the REGMAP (**B**), LANGUEDOC (**C**) and TOU-A (**D**) populations. Each dot represents the value of the content of a mineral element for one accession (yellow: ambient $CO_2$ (a$CO_2$, ~420 ppm), blue: elevated $CO_2$ (e$CO_2$, 900 ppm). N (% of dry weight), Fe ($\mu g.g^{-1}$ dry weight), Zn ($\mu g.g^{-1}$ dry weight), Cu ($\mu g.g^{-1}$ dry weight), Mg ($\mu g.g^{-1}$ dry weight), Mn ($\mu g.g^{-1}$ dry weight), Na ($\mu g.g^{-1}$ dry weight), C (% of dry weight). Asterisks indicate significant differences Paired Wilcoxson test; *, $p < 0.05$; **, $p < 0.005$; ***, $p < 0.0005$. ns; not significant.

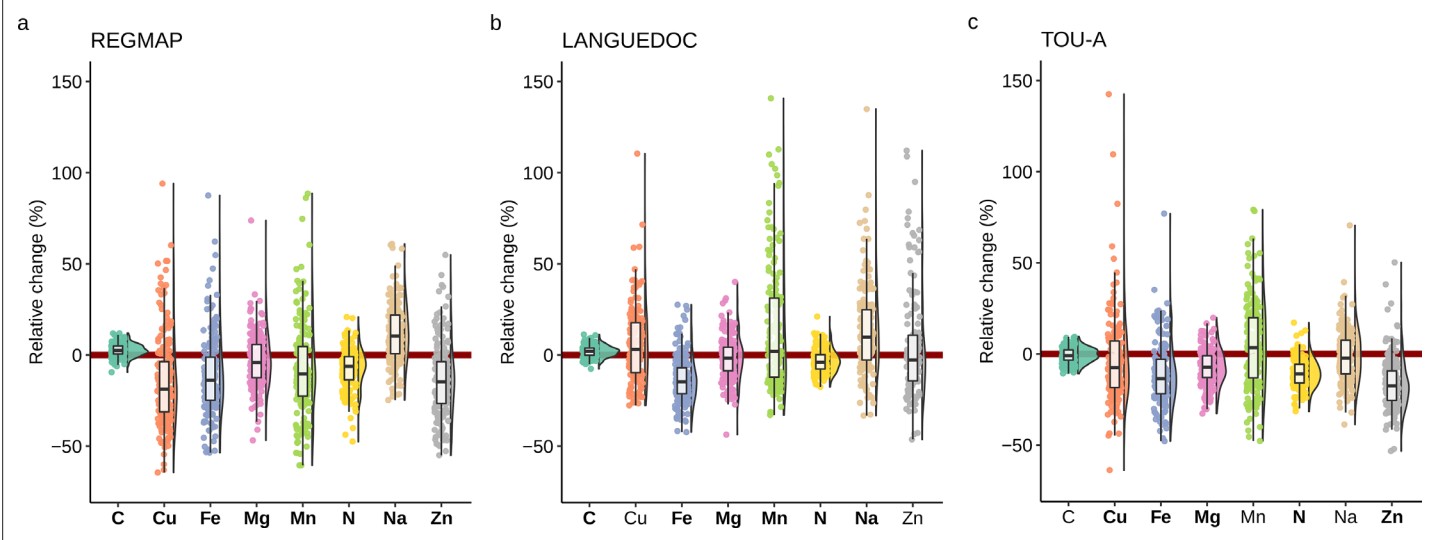

**Figure 2.** Elevated $CO_2$ leads to high phenotypic diversity of ionome response in *Arabidopsis thaliana*. Distributions of the relative change (%) of the content of 8 mineral elements between elevated $CO_2$ and ambient $CO_2$, in each population (A: REGMAP, B: LANGUEDOC, C: TOU-A). Each dot represents the value of the relative change of the content a mineral element for one accession. The name of the element appears in bold if the mean of the element in elevated $CO_2$ is significantly different from the mean of the element in ambient $CO_2$ (Paired Wilcoxon test, significance threshold of 0.05).

demonstrate that elevated $CO_2$ has on average a negative impact on the mineral content of natural genotypes of *A. thaliana* at the population-scale, whatever their geographic distribution.

## The ionome of *Arabidopsis thaliana* natural accessions displays a high range of phenotypic diversity in response to elevated $CO_2$

To explore the effect of elevated $CO_2$ in each accession, we calculated the relative change in nutrient composition of *A. thaliana* accessions from the three populations in response to elevated $CO_2$. In agreement with the results previously mentioned, we observed that the median relative change of most nutrient content at the population-level was negatively affected by elevated $CO_2$ (*Figure 2*). But the most striking observation was the genetic diversity of ionome response observed in these populations. Indeed, while most the natural accessions were negatively affected by elevated $CO_2$ (with a negative relative ratio of their nutrient content between ambient and elevated $CO_2$), a considerable number of accessions were rather not affected by elevated $CO_2$, or even positively affected, therefore showing an improved nutrient composition under elevated $CO_2$. For macronutrients like N, the relative change of concentration between ambient and elevated $CO_2$ varied from 20% to –50%, and for micronutrients like Cu, Fe or Zn, the relative change of concentration between ambient and elevated $CO_2$ varied from 100% to –60% (*Figure 2*). In addition, some differences among nutrients were observed between populations. For instance, a smaller dispersion of Fe relative change in the LANGUEDOC population, against a higher distribution of Mn relative change.

In order to explore the behavior of the different elements in response to elevated $CO_2$ and to observe the structure of phenotypic variation, we performed a principal component analysis (PCA) of the relative change in the 8 elements for the accessions from the three populations. The accessions from all populations seem to have globally similar responses to elevated $CO_2$, as suggested by the overlap of the three populations in the two first principal components (*Figure 3A*). The first component of the PCA described a clear antagonistic trend between C content and the change of other mineral elements (*Figure 3B*), suggesting that most of the variation between accessions in term of mineral response (almost 40%) could be driven by one or a few mechanisms resulting in an inverse variation between the whole ionome and C change (*Figure 3B*). Interestingly, the second component, explaining almost 15% of the variation among accessions in term of mineral response, was mainly driven jointly by change in N and C concentration. Altogether, these results show that there is a marked and large variability among accessions in their mineral concentration in response to elevated

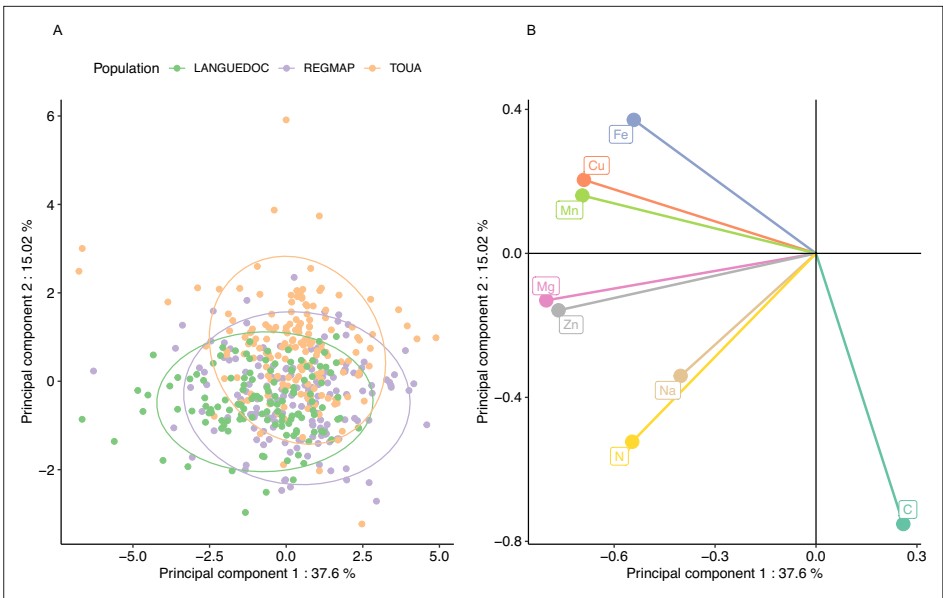

**Figure 3.** Elevated CO$_2$ results in a general pattern of ionome variation common to most accessions constituting natural populations of *Arabidopsis thaliana*. Principal Component Analysis (PCA) was performed using the variation of each element in response to elevated CO$_2$. (**A**) Natural accessions were positioned on the PCA and colored based on population. (**B**) Contribution of each element to the PCA axis.

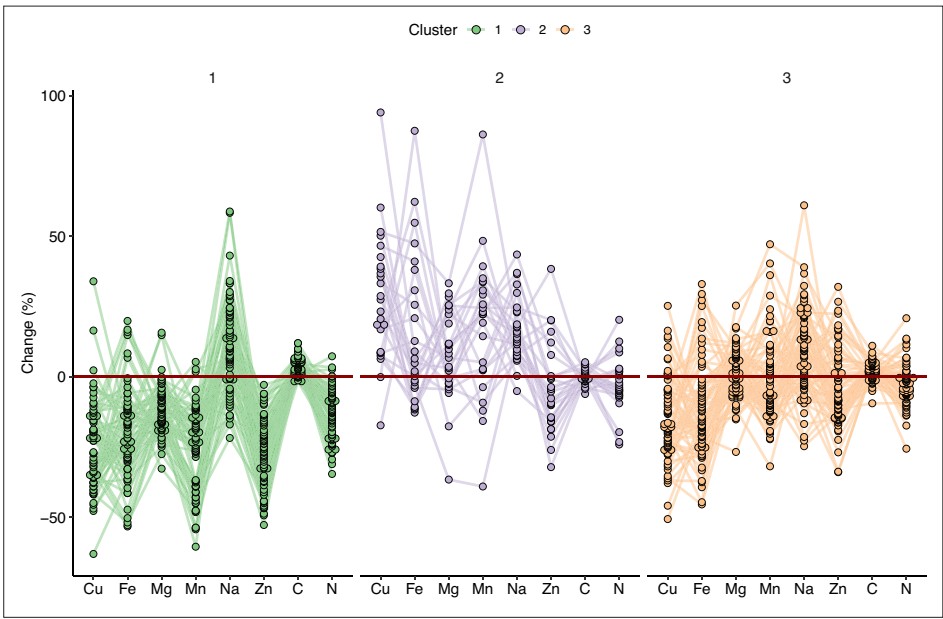

**Figure 4.** Variation in the response of the ionome to elevated CO$_2$ identifies contrasting subpopulations inside the REGMAP panel. K-means clustering was performed in the REGMAP accessions to identify different subpopulations. Each accession is represented by a dot, connected by a line between each element. Cluster 1:65 accessions. Cluster 2: 25 accessions. Cluster 3: 69 accessions.

The online version of this article includes the following figure supplement(s) for figure 4:

**Figure supplement 1.** Geographic distribution of accessions for each cluster identified within the REGMAP panel.

$CO_2$, illustrated by accessions negatively affected by elevated $CO_2$ and others positively affected by elevated $CO_2$.

## Genetic architecture of the ionome response to elevated $CO_2$, and identification of genetic determinants

In order to explore specific behavior of sub-populations, we clustered the accessions from the REGMAP panel via a k-means approach (**Figure 4**). This multivariate clustering resulted in the partitioning of accessions in three groups (**Supplementary file 1A**). Cluster 1 displayed the most negative pattern of ionome response to elevated $CO_2$. Inversely, accessions included in Cluster 2 displayed a globally positive response, with the highest relative change for almost all mineral elements, except for C content. These accessions did not appear to be clustered geographically with respect to their collection origin in the REGMAP panel (**Figure 4—figure supplement 1**), which is in line with the high genetic diversity of response to elevated $CO_2$ observed at smaller geographical scales (**Figure 2**). Finally, Cluster 3 displayed a resilient pattern, with accessions showing a globally attenuated response to elevated $CO_2$. Interestingly, the large phenotypic diversity of the ionome observed in the three populations in response to high $CO_2$, as well as the presence of contrasted subpopulations in the REGMAP panel, suggests the presence of genetic determinants associated with this response.

We ran Genome-Wide Association (GWA) mapping to describe the genetic architecture of the ionome response to elevated $CO_2$, and to fine-map candidate genes underlying the detected quantitative trait loci (QTLs). We focused here on the phenotypic data collected on the REGMAP population, and used the sequencing data available for this population (**Arouisse et al., 2020**). We included in this analysis the level of each mineral under ambient and under elevated $CO_2$, as well as the relative change between ambient and elevated $CO_2$ for each element. We also included a trait corresponding for each accession to the coordinate on the first and on the second PCA axes (PCA1 and PCA2) explaining collectively more than 50% of ionomic variation (**Figure 3**). Therefore, these values correspond to traits driving and summarizing a large part of the ionome variation under elevated $CO_2$. This resulted as a whole in running GWA mapping on 30 different single-trait GWAS. The overall approach was first validated by observing expected results for traits phenotyped under ambient $CO_2$. For instance, we observed a very strong peak for the Na content at the locus of the *HKT1* gene (**Figure 5—figure supplement 1A**), which is known to be involved in the natural genetic variation of Na content in *Arabidopsis thaliana* (**Baxter et al., 2010**), or a strong peak for the N content at the locus of the *NIA1* gene (**Figure 5—figure supplement 1B**), encoding for an isoform of the nitrate reductase required for the first step of nitrate reduction and associated with natural genetic variation of N content in *A. thaliana* (**North et al., 2009**).

GWA mapping revealed a polygenic architecture for each phenotypic trait, although its complexity largely differs among traits (**Figure 5**). For instance, very few and neat peaks of association were detected Na and Mn content under elevated CO2, or of Fe and Cu relative change between ambient and elevated $CO_2$ (**Figure 5—figure supplements 2 and 3**). On the other hand, a more complex genetic architecture with the detection of a large number of QTLs was observed for traits related to N or C content (**Figure 5—figure supplements 2 and 3**). For each of the traits that have been analyzed under elevated $CO_2$ or corresponding to the relative change of their content between ambient and elevated $CO_2$, we isolated the 50 SNPs with the most significant p-value, hereafter named top SNPs (**Figure 5A** and **Supplementary file 1B and C**). In order to identify the overlap between the genetic architecture of each trait, we looked whether some of the top SNPs were shared among traits. While the large majority of SNPs were specific to one trait, 30 and 21 SNPs were shared between two traits for the content under elevated $CO_2$ or for the relative change between ambient and elevated $CO_2$, respectively (**Figure 5B** and **Supplementary file 1B and C**). In addition, 8 and 2 SNPs were shared between three traits for the content under elevated $CO_2$ or for the relative change between ambient and elevated $CO_2$, respectively (**Figure 5B** and **Supplementary file 1B and C**). Most of the shared SNPs were associated with micronutrients (Fe, Mn, Zn, and Mg content) and with N and/or with the first component of the PCA axis. An interesting QTL located on chromosome 1 was notably associated with 6 traits, displaying SNPs shared between Mn, Zn, and N relative change and SNPs shared between Mn, N, and PC1 content under elevated $CO_2$ (**Figure 5A** and **Supplementary file 1B and C**). Another QTL located on chromosome 3 encompasses SNPs shared between Fe, Zn, and PC1 content under elevated $CO_2$ (**Figure 5A** and **Supplementary file 1B and C**).

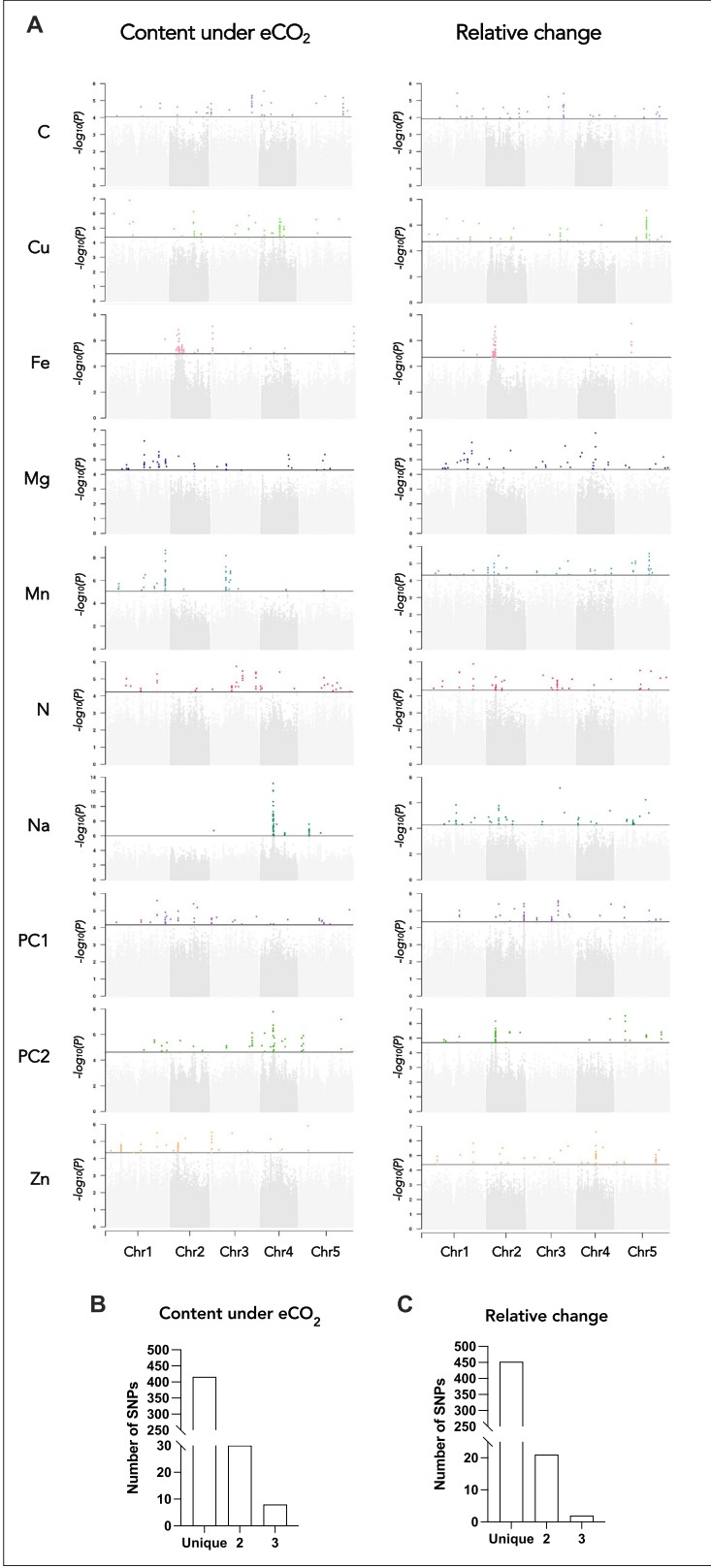

**Figure 5.** Genetic architecture of the response of the ionome to elevated $CO_2$ in the REGMAP panel of *Arabidopsis thaliana*. (**A**) Manhattan plots for the content of eight mineral elements under elevated $CO_2$, or for the relative change of the content of mineral elements between elevated $CO_2$ and ambient $CO_2$. For each Manhattan plot, SNPs with the 50 most significant p-value, located above the horizontal line, are colored. Bar plots showing

*Figure 5 continued on next page*

*Figure 5 continued*

the number of SNPs identified by GWAs for traits under elevated $CO_2$ (**B**) or for the relative change of the content of mineral elements between elevated $CO_2$ and ambient $CO_2$ (**C**) that are unique to one element or shared between two or three traits.

The online version of this article includes the following figure supplement(s) for figure 5:

**Figure supplement 1.** Snapshots of the Manhattan plots of Na content (**A**) and N content (**B**) under ambient $CO_2$ in the REGMAP panel.

**Figure supplement 2.** Qqplots from the GWAs corresponding to data for the level of each mineral under ambient and under elevated $CO_2$, as well as for the percentage of change between ambient and elevated $CO_2$ for each element.

**Figure supplement 3.** Manhattan plots made from the GWAs corresponding to data for the level of each mineral under elevated $CO_2$, as well as for the percentage of change between ambient and elevated $CO_2$ for each element.

We next identified for each trait a list of the genes located at ±25 kb from the top 50 SNPs, which corresponds to the rough estimate of the decay of linkage disequilibrium identified in *A. thaliana* at the worldwide scale (***Kim et al., 2007***). This resulted in a list of genes for each element, ranging from 154 to 422 genes depending on the element (***Supplementary file 1B and C***). Among others, several genes associated with top 50 SNPs were identified as obvious candidates of the effect of elevated $CO_2$ on plant nutrition and ionome content. This was the case of *ZINC INDUCED FACILITATOR 1* (*ZIF1*, *AT5G13740*) and *ZIF-LIKE1* (*AT5G13750*), linked with SNPs identified for Zn content under elevated $CO_2$, and involved Zn sequestration mechanisms (***Lee et al., 2021***). We also noticed the link between SNPs identified for Zn relative change and *TIP2;2* (*AT4G17340*), known to be involved in Zn root-to-shoot translocation (***Wang et al., 2022***). Concerning N relative change, some of the top 50 SNPs were linked to *ASN1* (*AT3G47340*), which is an actor of N status and remobilization (***Lam et al., 2003***; ***Gaufichon et al., 2017***). Some of the top 50 SNPs identified for Fe relative change were linked to *MCO2* (*AT5G21100*) and *MCO3* (*AT5G21105*) genes, which have been recently characterized as actors of the regulation of Fe homeostasis (***Brun et al., 2022***). Finally, it is interesting to note that the QTL located on chromosome 3 mentioned above displaying significant shared SNPs identified for Fe, Zn, and PC1 content under elevated $CO_2$ was associated among other genes with *ISU2* (*AT3G01020*), coding for one of the Fe-S clusters in *Arabidopsis thaliana*, which are known to be essential for photosynthesis and metabolism (***Balk and Schaedler, 2014***). Altogether, this demonstrated that genes identified through this approach represent a large and valuable reservoir of candidates to study and to counteract the effect of elevated $CO_2$ on plant nutrition and ionome content.

To analyze how these genes identified by GWA mapping are regulated by elevated $CO_2$, we performed RNA-seq from shoots and roots grown under ambient and elevated $CO_2$. Differentially expressed genes (DEG) associated to the effect of elevated $CO_2$ were identified from shoots and roots (***Supplementary file 1D***). We compared the list of shoots or roots elevated $CO_2$-DEG with the list of genes identified by GWA mapping for each element, which resulted in a list of 182 genes identified by GWA mapping and differentially regulated by elevated $CO_2$ in shoot or in roots (***Supplementary file 1E***), making them relevant candidates to be involved in the response of the mineral composition of plants to elevated $CO_2$. Most of these genes were deregulated by elevated $CO_2$ in shoot (***Figure 6A and B***). In shoot or in roots, these genes mainly showed an association with C-, Mg-, or Zn-related traits (***Figure 6A and B***). Several of these genes, identified by GWA mapping and whose expression is deregulated in response to high $CO_2$, were known for their role in nutrient homeostasis. This was the case for the *ASN1* and *DUR3* genes, encoding an asparagine synthase and a urea transporter involved in N metabolism and remobilization, both associated here with a N-related peak of association, and whose expression is modulated by high $CO_2$ in leaves (***Figure 6C***). We also observed in the leaves an interesting profile for several genes related to C metabolism and photosynthesis. This was the case for the *BGP3* gene, involved in chloroplast development, or for the carbonic anhydrase *CA1*, both showing a decreased expression in response to high $CO_2$ and both associated with a peak in C-related GWA mapping under elevated $CO_2$ (***Figure 6C***). In roots, the gene most deregulated in response to high $CO_2$ was *AT1G64710*, encoding a GroES-type alcohol dehydrogenase, which interestingly is also deregulated in leaves (***Figure 6D***). We also observed in the roots that the expression of the *TIP2;2*

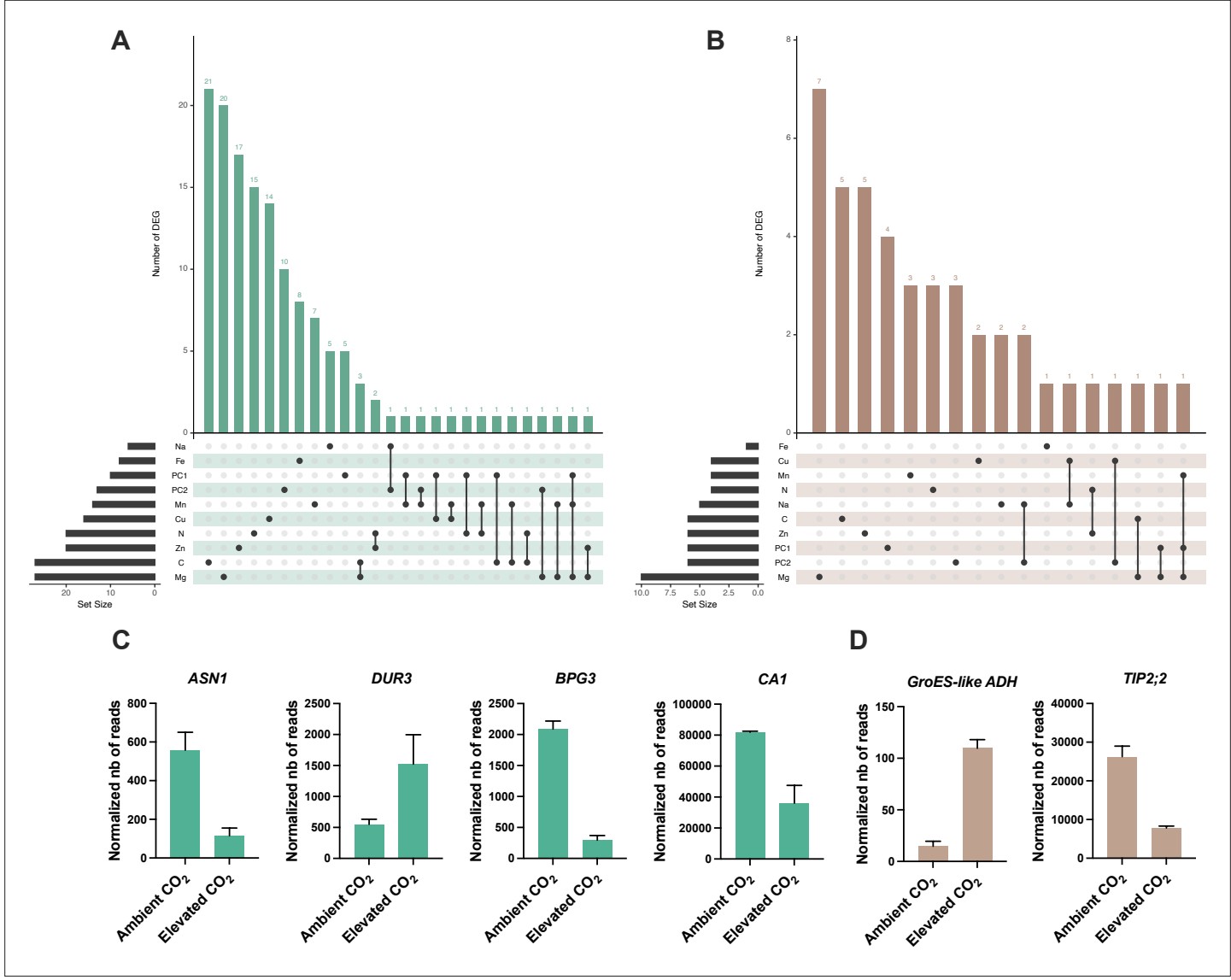

**Figure 6.** Identification of genes detected by GWA mapping and differentially regulated by elevated $CO_2$. Intersection between elevated $CO_2$-DEG in shoot (**A**) or root (**B**) and genes identified by GWA mapping. UpSet plots display the number of elevated $CO_2$-DEG that are associated to a locus identified for the content or the relative change of one or several mineral elements under elevated $CO_2$. Illustration of the pattern of elevated $CO_2$-DEG in shoot (**C**) or root (**D**) also identified by GWA mapping.

gene, associated here with a peak detected for Zn relative change GWA mapping, was deregulated in response to elevated $CO_2$ (**Figure 6D**).

To go further, we selected one of the association peaks identified by GWA mapping, and sought to functionally validate the importance of this QTL in response to elevated $CO_2$, in order to demonstrate the value of our data set and GWA mapping analyses. To do so, we selected an association peak located on chromosome 4 and associated with Zn relative change (**Figure 7A**). More precisely, this association peak displayed the SNPs with the most significant *p*-values and more largely three SNPs that fell into the top 10 SNPs of the trait corresponding to the Zn relative change between ambient and elevated $CO_2$. The SNPs corresponding to the alternative alleles were associated to an increase of Zn content under elevated $CO_2$ (**Figure 7B**). These SNPs are located very close to the *TIP2;2* (*AT4G17340*) gene, which has been recently characterized as an actor of Zn root-to-shoot translocation (**Wang et al., 2022**). We thus selected a set of accessions from haplotype 0 (reduced Zn content under elevated $CO_2$) or haplotype 1 (increased Zn content under elevated $CO_2$), and analyzed *TIP2;2* expression in the roots under ambient and elevated $CO_2$. This analysis revealed a haplotype-specific difference in *TIP2;2*

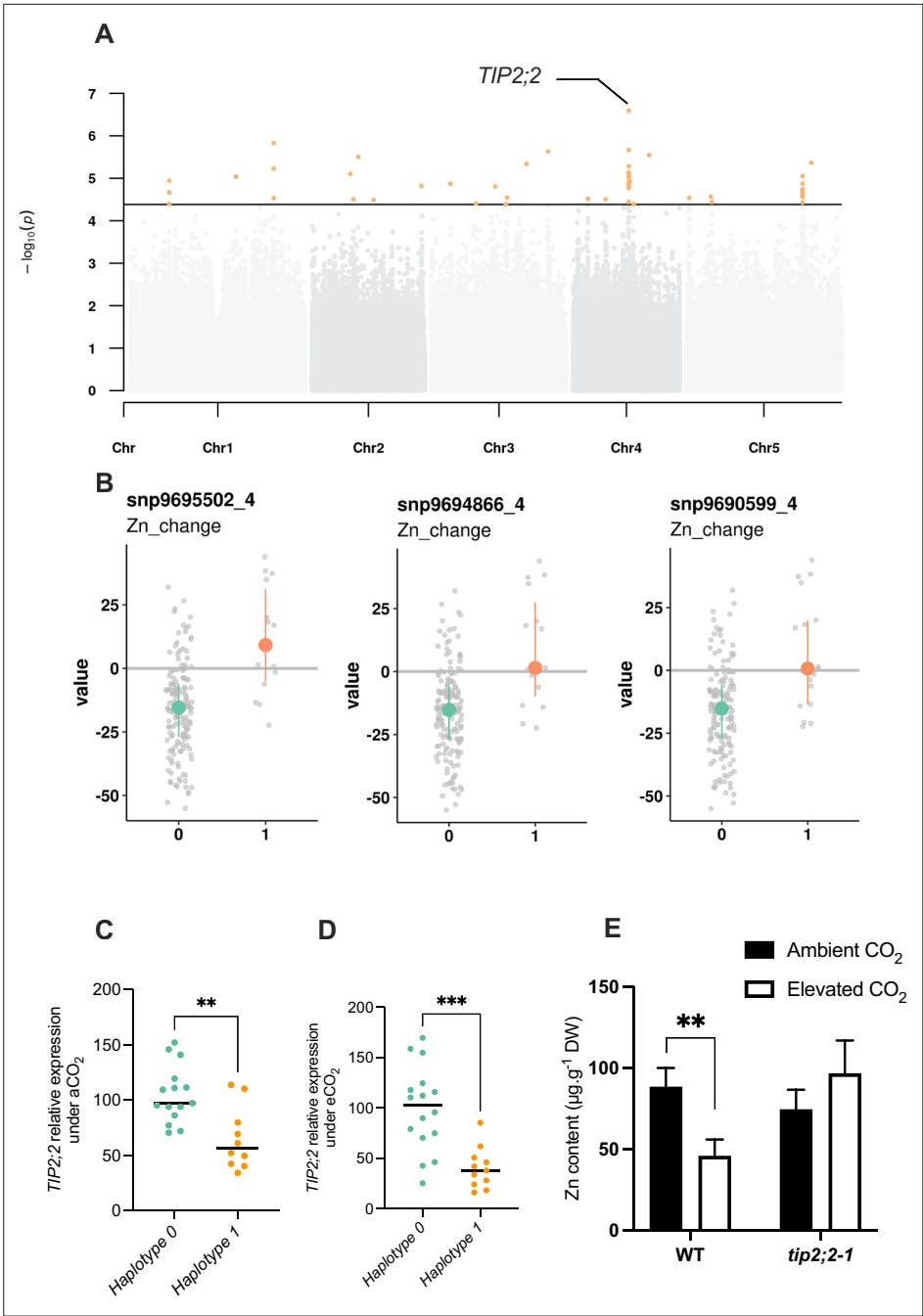

**Figure 7.** Natural variation of the *TIP2;2* gene is associated with improved responses of Zn content to elevated $CO_2$. (**A**) Manhattan plot of the relative change of Zn content between elevated $CO_2$ and ambient $CO_2$ showing the presence of a peak closed to the *TIP2;2* locus. (**B**) Comparison of haplotypes and their relative change of Zn content between elevated $CO_2$ and ambient $CO_2$. Three SNPs located at the *TIP2;2* locus are associated to an improvement of Zn content under elevated $CO_2$ for accessions that possess them (haplotype 1) compared to the rest of the population (haplotype 0). (**C, D**) Relative expression of *TIP2;2* in the roots under ambient (**C**) or elevated (**D**) $CO_2$ for accessions belonging to haplotype 0 or haplotype 1. Relative expression levels were calculated based on *UBQ10* as internal control. Horizontal black line represented the median of each group of haplotypes. \*\*\*$p<0.001$, \*\*$p<0.01$, unpaired Mann-Whitney test. (**E**) Shoot Zn content under ambient or elevated $CO_2$ for WT (Columbia) and *tip2;2–1* mutant. Data are presented as the mean (with SD) of 5 and 6 biological repeats for the WT and *tip2;2–1*, respectively. \*\*$p<0.01$, unpaired Mann-Whitney test.

expression: under ambient and elevated $CO_2$, accessions from haplotype 1 (correlated with a higher Zn content in the shoot (*Figure 7B*)) show a reduced *TIP2;2* expression in the roots compared to those from haplotype 1 (*Figure 7C and D*), with a reduction being more pronounced under elevated $CO_2$. To test the effect of *TIP2;2* expression of Zn content under elevated $CO_2$, we used the *tip2;2–1* knock-out mutant and compared its Zn content under ambient and elevated $CO_2$ to this of the WT. We observed that the *tip2;2–1* mutant line did not present any decrease in Zn shoot content in response to elevated $CO_2$, in opposition to what is observed for the WT (*Figure 7E*). In the *tip2;2–1* mutant, Zn content under elevated $CO_2$ was even slightly higher than under ambient $CO_2$. Altogether, these results demonstrated that these data sets generated in this study and the associated analyses are a valuable resource to identify genes and associated mechanisms involved in the effect of elevated $CO_2$ on the mineral composition of plants.

## Discussion

### The natural variation of ionome response to elevated $CO_2$ in *Arabidopsis thaliana* displays a high degree of genetic variation

In the present work, we analyzed the diversity of leaf ionome response to elevated $CO_2$ present in the natural variation of *Arabidopsis thaliana*. In agreement with several other phenotypic traits related to phenology and disease resistance (*Brachi et al., 2013*; *Huard-Chauveau et al., 2013*; *Roux and Frachon, 2022*), we observed a wide range of responses at complementary geographical scales, from accessions with an ionome negatively affected by high $CO_2$ to accessions with an ionome benefiting from high $CO_2$. This confirms for the first time on large and complementary sets of natural genotypes what has been observed by meta-analysis on isolated groups of plants worldwide (*Loladze, 2014*; *Myers et al., 2014*). The global analysis of the distribution of each mineral element studied suggests firstly a trend where the whole ionome would evolve in a unified manner in response to high $CO_2$, and in an opposite manner to C. This is in line with a number of studies that have proposed that the accumulation of carbohydrates due to the stimulation of photosynthesis by high $CO_2$ would be the cause of the decrease in plant mineral composition (*Ainsworth and Long, 2005*; *Thompson et al., 2017*; *Dusenge et al., 2019*; *Tausz-Posch et al., 2020*). However, the reading of the genetic architecture performed here by a genome-wide association genetics approach suggests that the majority of the genetic mechanisms underlying the negative effect of elevated $CO_2$ on the ionome are specific to each mineral element. Some specific cases, such as the QTL detected on chromosome 1 and associated with the natural genetic variation of 6 traits among the 20 considered, will certainly deserve a more in-depth analysis.

By clustering globally distributed accessions according to their ionome sensitivity to high $CO_2$, we were able to observe that the geographic origin of the accessions likely did not determine their response to $CO_2$. This suggests that inherent genetic factors, more than those due to local adaption, direct the response of plants to elevated $CO_2$. This seems consistent since the $CO_2$ elevation applied here to natural *Arabidopsis thaliana* variants does not correspond to any environment experienced by plants yet, at least for several tens of millions of years (*Pearson and Palmer, 2000*; *Lüthi et al., 2008*). In this context of brutal and highly impactful environmental change, the presence of cryptic genetic variation often explains the appearance of relatively rapid adaptive mechanisms (*Pauls et al., 2013*; *Cortés and López-Hernández, 2021*). Although not formally tested here, it would be interesting to examine whether the variation in the ionome in response to elevated $CO_2$ shows evidence of cryptic variation. In any case, the presence of high phenotypic diversity in these natural populations of *A. thaliana* demonstrates very clearly the possibility of taking advantage of this genetic variation to understand and alleviate the negative response of plant mineral composition to high $CO_2$.

### GWA mapping of ionome variation under elevated $CO_2$ identified a large number of genes to understand and mitigate the negative effect of high $CO_2$ on plant mineral composition

In order to understand the genetic mechanisms underlying the effect of high $CO_2$ on plant mineral composition, and to enable future breeding approaches, we adopted an association genetics approach. This led to the identification of a large number of candidate genes associated to the variation of nutrients under elevated $CO_2$. Several genes in this list can easily attract attention. In

particular, we can note the identification of *ASN1* and *DUR3* genes in two of the loci associated with N content variation under elevated $CO_2$. *ASN1*, and to a lesser extent *DUR3*, play an important role in the remobilization and the reallocation of N within the plant, and their manipulation can lead to variation in N use efficiency (*Lam et al., 2003*; *Bohner et al., 2015*; *Gaufichon et al., 2017*). This is interesting because for the moment, root N uptake and N assimilation seemed to be the key targets of the negative effect of high $CO_2$ on plant N content (*Bloom et al., 2010*; *Cassan et al., 2023*), but these results suggest that remobilization of N may also be involved. We also identified the *CA1* gene, coding for a carbonic anhydrase, in the vicinity of a QTL associated with C variation under high $CO_2$. *CA1* is involved in the regulation of stomatal opening by elevated $CO_2$ (*Hu et al., 2015*), and the β carbonic anhydrase family of which *CA1* belongs is involved in the regulation of photosynthetic efficiency, although *CA1* shows no significant effect under standard conditions (*Sharma et al., 2023* ). It would be therefore interesting to assess the role of *CA1* natural genetic variation under elevated $CO_2$. If *CA1* regulates the C variation of the ionome under elevated $CO_2$, this could, according to our observations, significantly influence the global mineral composition of plants. Interestingly, the genes identified by GWA mapping in the ionome response to high $CO_2$, including those mentioned above, showed substantial variation at the gene expression level. We ended this study with the functional validation of an association peak identified by GWA mapping for the relative change of Zn content between ambient and elevated $CO_2$. Zn is an essential element for a large number of metabolic processes in humans, and Zn deficiency, found in up to one third of the world's population, leads to severe health problems. We demonstrated that *TIP2;2* gene expression varied in a haplotype-specific manner, in both ambient and elevated $CO_2$. In parallel to this, we show that loss of *TIP2;2* expression using a knock-out mutant can abolish the Zn decrease observed under high $CO_2$. It therefore seems that variation in *TIP* expression is associated with the effect that atmospheric $CO_2$ can have on zinc levels. A recent study demonstrated that TIP2;2 was responsible for Zn retention in the roots (*Wang et al., 2022*). It therefore seems consistent that natural accessions with the lowest expression levels of this gene are those with the highest Zn content in aerial parts, due to low retention in their roots. This example illustrates the potential of the resource we have generated here towards the identification of genes involved in the variation of leaf ionome in response to rising $CO_2$, and towards the characterization of the associated mechanisms. The understanding of these mechanisms represents a considerable challenge in view of the current rise in atmospheric $CO_2$, which might be useful to the coming breeding programs for crops adapted to a forthcoming $CO_2$-rich atmosphere (*Shahzad and Rouached, 2022*). It must however be recalled that leaf ionome is not always correlated with seed ionome, in particular in *Arabidopsis thaliana* accessions (*Campos et al., 2021*). Therefore, the effect of elevated $CO_2$ on seed ionome, certainly in major crops, must be the next target to analyze in order to pave the way for the development of nutritious crops adapted to elevated $CO_2$.

## Methods
### Plant material
A subset of the REGMAP panel, the LANGUEDOC panel, and the TOU-A panel were used in this study. These populations were previously described here (*Horton et al., 2012*; *Brachi et al., 2013*; *Frachon et al., 2017*). These populations were grown on Jiffy-7 peat pellets (Jiffy Products International, NL) under ambient (~420 ppm) or elevated (900 ppm) $CO_2$ in the growth chambers of the Microcosms experimental platform at the Montpellier European Ecotron CNRS. Five replicates of each accession were randomly distributed in the growth chambers. The concentration of 900 ppm of $CO_2$ was chosen as current $CO_2$ emissions align with the IPCC's RCP8.5 model, which predicts a CO2 concentration of around 900 ppm in 2100. Growth conditions were 6 hr/22 hr light (22 °C) / dark (20°) photoperiod, with 200 µmol m$^{-2}$ s$^{-1}$ light intensity and 65% of hygrometry. Plants were watered twice a week with a growth solution containing $KH_2PO_4$ 1 mM, $MgSO_4$ 1 mM, $K_2SO_4$ 250 µM, $CaCl_2$ 250 µM, Na-Fe-EDTA 100 µM, $KNO_3$ 10 mM, KCl 50 µM, $H_3BO_3$ 30 µM, $MnSO_4$ 5 µM, $ZnSO_4$ 1 µM, $CuSO_4$ 1 µM, $(NH_4)_6Mo_7O_{24}$ 0,1 µM, as described by *Gansel et al., 2001*. The entire rosettes were collected three weeks after sowing. The *tip2;2–1* mutant line corresponds to the *SALK_152463* allele (*Wang et al., 2022*).

## Ionome analysis

From three to five replicates per accession were pooled and used for each ionome analysis. Total C and N content was obtained from dried shoot tissue using an Elementar Pyrocube analyzer. Cu, Fe, Mg, Mn, Na, and Zn content was obtained from dry shoot tissue mixed with 750 µl of nitric acid (65% [v/v]) and 250 µl of hydrogen peroxide (30% [v/v]). After one night at room temperature, samples were mineralized at 85 °C during 24 hr. Once mineralized, 4 ml of milliQ water was added to each sample. Mineral contents present in the samples were then measured by microwave plasma atomic emission spectroscopy (MP-AES, Agilent Technologies).

## Removal of outlier observations

Prior to GWAS and multivariate analyses such as PCA or clustering, mineral composition measures were pre-processed to remove technical outliers. For a given element and $CO_2$ condition, the values positioned more than 5 median absolute deviations away from the median were removed from the dataset. The number of outliers removed from each dataset is indicated in *Supplementary file 1G*.

## PCA and clustering

Principal Component Analysis was performed using the R *ade4* package after the prior scaling of the variables to a z-score. Clustering of the REGMAP panel based on the relative changes of the mineral composition of each accession has been done using a k-means clustering with the R *kmeans* function. For this step, the variables were also scaled to a z-score. The number of clusters in the k-means algorithm was chosen by the elbow method on the criteria of cluster homogeneity (within-sum of squares).

## GWAs

Genome-Wide Association mapping was performed using the R *statgenGWAs* package. Genotype data was prepared using the *codeMarkers* function, removing duplicated SNPs and filtering for a minimum allele relative frequency of 0.04. Associations were performed by the *runSingleTraitGwas* function, that implements the EMMA algorithm. Population structure was modeled via a kinship matrix built from the Astle method. Manhattan plots were drawn using the *manPlotFast* function of the *ramwas* R package.

## RNA-seq experiments

Plants from the Columbia accession were grown in hydroponics to have access to the roots in addition to the shoot, as previously described in *Cassan et al., 2023*. Shoot or root from five plants were pooled into one biological replicate, flash frozen in liquid nitrogen, and stored at –80°C. RNA of three biological replicates were extracted from shoot or root tissues using Direct-zol RNA Miniprep (Zymo Research, CA, USA), according to the manufacturer recommendations. RNA-sequencing libraries were done from shoot or root total RNA using standard RNA-Seq protocol method (Poly-A selection for mRNA species) by the Novogene company. RNA-sequencing was performed using Illumina technology on a NovaSeq6000 system providing PE150 reads. The quality control and adapter trimming of raw paired-end fastq files was done with *fastp* and its default parameters. Mapping to the TAIR10 reference genome was performed with STAR, and using the following options:

- `--outSAMtype BAM SortedByCoordinate`
- `--outFilterMismatchNmax 1`
- `--outFilterMismatchNoverLmax 0.15`
- `--alignIntronMin 30`
- `--alignIntronMax 5000`

Quantification of the bam files against the TAIR10 GFF3 annotation file was done using *htseq*-count with options:

- `-f bam --type gene -r pos`
- `--idattr=Name --stranded=no`

Normalization and differential expression were performed using *DIANE* R package (*Cassan et al., 2021*), with no fold change constraint, and an adjusted p-value threshold (FDR) of 0.05. Lowly

expressed genes with an average value across conditions under 25 reads were excluded from the analysis.

## Quantitative real-time PCR

Plants were grown in hydroponics to have access to the roots, as previously described in *Cassan et al., 2023*. Root tissue from 5five plants were pooled into one biological replicate, flash frozen in liquid nitrogen, and stored at –80 °C. RNA were extracted from shoot or root tissues using TRIZOL (Invitrogen, USA), according to the manufacturer recommendations, and DNAse treated using RQ1 (Promega, USA). Reverse transcription was achieved from 1 µg of total RNA with M-MLV reverse transcriptase (RNase H minus, Point Mutant, Promega, USA) using an anchored oligo(dT)20 primer. Accumulation of transcripts was measured by qRT-PCR (LightCycler 480, Roche Diagnostics, USA) using the SYBR Premix Ex TaqTM (TaKaRa, Japan). Gene expression was normalized using *UBQ10* and *ACT2* as internal standards. Results are presented as the expression relative to *UBQ10*. Sequences of primers used in RT-qPCR for gene expression analysis are listed in *Supplementary file 1F*.

## Acknowledgements

This work was supported by the I-Site Montpellier Université d'Excellence (MUSE; project ECO2THREATS), the CNRS through the Mission for Transversal and Interdisciplinary Initiatives (MITI) 80 PRIME program, and the Plant Biology and Breeding department of INRAE. O.C. was recipient of a PhD fellowship from the CNRS. T.M. was recipient of a PhD fellowship from INRAE and Région Occitanie. We thank Jiping Liu from Cornell University for the gift of *tip2;2–1* mutant line. This study benefited from the CNRS human and technical resources allocated to the Ecotron Research Infrastructure and from the state allocation "Investissement d'Avenir" AnaEE France ANR-11-INBS-0001. A CC-BY public copyright license has been applied by the authors to the present document.

## Additional information

### Funding

| Funder | Grant reference number | Author |
| --- | --- | --- |
| Centre National de la Recherche Scientifique | | Oceane Cassan<br>Sophie Lebre<br>Antoine Martin |
| Montpellier Université d'Excellence | | Lea-Lou Pimpare<br>Cecile Fizames<br>Sebastien Devidal<br>Alexandru Milcu<br>Alain Gojon<br>Antoine Martin |

The funders had no role in study design, data collection and interpretation, or the decision to submit the work for publication.

### Author contributions

Oceane Cassan, Data curation, Formal analysis, Visualization, Methodology, Writing - original draft, Writing - review and editing; Lea-Lou Pimpare, Sebastien Devidal, Methodology; Timothy Mozzanino, Formal analysis, Validation, Methodology; Cecile Fizames, Formal analysis, Visualization; Fabrice Roux, Resources, Writing - original draft, Writing - review and editing; Alexandru Milcu, Conceptualization, Funding acquisition, Writing - original draft, Writing - review and editing; Sophie Lebre, Conceptualization, Funding acquisition, Investigation, Methodology; Alain Gojon, Conceptualization, Funding acquisition, Writing - original draft, Project administration, Writing - review and editing; Antoine Martin, Conceptualization, Supervision, Funding acquisition, Investigation, Methodology, Writing - original draft, Project administration, Writing - review and editing

### Author ORCIDs

Antoine Martin  http://orcid.org/0000-0002-6956-2904

Reviewer #1 (Public Review): https://doi.org/10.7554/eLife.90170.3.sa1
Reviewer #2 (Public Review): https://doi.org/10.7554/eLife.90170.3.sa2
Author response https://doi.org/10.7554/eLife.90170.3.sa3

## Additional files

### Supplementary files

• Supplementary file 1. Clustering of the REGMAP population and expression data in response to $CO_2$. (**a**) List of accessions per clusters identified by k-means approach in the REGMAP panel. (**b**) List of top 50 SNPs identified by GWAs for the content of each element under $eCO_2$. "snp": coordinates of the SNP identified. "shared": number of traits for which the SNP is identified in the top 50 list. "element": element for which the SNP has been identified in the top 50 list. "chrom": chromosome number on which the SNP is positionned. "position": position of the SNP in the chromosome. "AGIs": list of genes located at -/+25 kb from the SNP. (**c**) List of top 50 SNPs identified by GWAs for the relative change of each element between $eCO_2$ and $aCO_2$. "snp": coordinates of the SNP identified. "shared": number of traits for which the SNP is identified in the top 50 list. "element": element for which the SNP has been identified in the top 50 list. "chrom": chromosome number on which the SNP is positionned. "position": position of the SNP in the chromosome. "AGIs": list of genes located at -/+25 kb from the SNP. (**d**) List of genes differentially expressed by eCO2 in shoot or root. (**e**) List of genes identified by GWAs for each element that are differentially expressed by eCO2 in shoot or root. (**f**) List of primers used for qRT-PCR. (**g**) Number of outliers removed from each dataset.

• MDAR checklist

### Data availability

Data and R notebooks containing the analyses performed in this article can be found at https://src. koda.cnrs.fr/groups/ipsim/sirene-team (copy archived at *Cassan et al., 2024*). RNA-seq data generated for this study are available at https://www.ebi.ac.uk/biostudies/arrayexpress/studies using the accession no E-MTAB-13661.

The following dataset was generated:

| Author(s) | Year | Dataset title | Dataset URL | Database and Identifier |
|---|---|---|---|---|
| Cassan O, Pimparé L-L, Mozzanino T, Fizames C, Devidal S, Roux F, Milcu A, Lèbre S, Gojon A, Martin A | 2023 | RNA-seq of *Arabidopsis thaliana* root and shoot in response to elevated atmospheric CO2 | https://www.ebi. ac.uk/biostudies/ arrayexpress/studies/ E-MTAB-13661? query=E-MTAB-13661 | EBI Array Express, E-MTAB-13661 |

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
