## [Editor Report · eLife assessment]

This paper provides **useful** information about how the ionome of *Arabidopsis thaliana* adapts to very high CO2-levels, backed up by **solid** evidence and carefully designed studies. The work will be of interest to anyone studying natural genetic variation as well as the response of plants to altered CO2 levels in the atmosphere.

---

## [Referee Report · Reviewer #1 (Public Review)]

This study offers good evidence pointing to a genetic basis for *Arabidopsis thaliana's* response to elevated CO2 (eCO2) levels and its subsequent impact on the leaf ionome. The natural variation analyses in the study support the hypothesis that genetic factors, rather than local adaptation, guide the influence of eCO2 on the ionome of rosette leaves in Arabidopsis.

Comments on current version:

I appreciate the revisions and the effort the authors have made.

Most of the abstract now accurately reflects the results and methods. It would be nice to have a few more technical details in the abstract, such as:

* What was the CO2 level?

* Which gene was identified?

I still have a problem with this sentence:

"The elevation of atmospheric CO2 leads to a decline in plant mineral content, which might pose a significant threat to food security in the coming decades."

The authors provide a wide range of published studies that support this statement. I fully agree that this is what the literature suggests. However, I think the literature has asked the wrong question.

In general, these studies addressed the question: Given no time for adaptation, do plants grown under high CO2 have a different mineral composition? The answer is yes.

But a more important question is: Can plants and food crops adapt in time? I believe the strength of this study is that it tests this, and it suggests that the answer is yes. I also think there is a lot of unpublished results and greenhouse breeding success that supports the contention that most plants can adapt to the CO2.

"The artificial elevation of atmospheric CO2 leads to a physiological response and decline in plant mineral content, which might pose a significant threat to food security in the coming decades if plants cannot adapt."

It needs to be made clear throughout the paper when high CO2 levels lead to low mineral composition. These are all artificial manipulations without allowing the plants to adapt to the new environment.

"The elevation of atmospheric CO2 concentration leads to a decline in the mineral composition of C3 plants (Gojon et al., 2023)." - this is well supported in artificial environments.

Do wild plants have fewer minerals in their leaves today compared to plants in 1950? This would be great evidence and framing for this experiment.

Crop plants having lower nitrogen and different mineral compositions over time is substantially a product of breeders initially increasing inputs and then, over the last decade, selecting for higher input efficiency.

At the end of the introduction or the beginning of the results, please define why the CO2 level was chosen and its context as being at the high end of current predictions.

"According to the literature, this results in a 20-25% reduction in vitamin C or lycopene and requires a significantly higher nitrogen and water intake to reach expected sugar levels (Doddrell H (2023), Horticulture Research). In addition, the negative effect of elevated CO2 on tomato nutrient content seems to have significant repercussions on nutrition-health properties (Boufeldja (2023), Molecules)."

Thank you for sharing these reviews. These suggest to me that breeders favored the 80% yield bump over other traits. Either there was no breeding, or the breeding focused on other traits. It is important to mention that breeders should include mineral nutrition in their selection index while they maximize yield. Simpler breeding strategies can sometimes heavily favor one trait over others, but cattle breeders today regularly use selection indices that incorporate weights for two dozen traits.

This study provides nice evidence that an annual weed species is likely to be able to adapt easily to high eCO2. Whether perennial species will be able to adapt in time is clearly a topic that needs to be investigated.

---

## [Referee Report · Reviewer #2 (Public Review)]

The research uses a large collection of *Arabidopsis thaliana* accessions from various geographic scales to investigate the natural genetic variation underlying the response of ionome (elemental) composition to elevated CO2 (eCO2), a concern for future food security. While most accessions show a decrease in elemental accumulation, the authors demonstrate a wide variety of responses to eCO2 across the diversity of Arabidopsis, including lines that increase elemental content in eCO2. The demonstration of genetic diversity in eCO2 response is a significant contribution to our understanding of this important phenomenon.

Comments on revised version:

The authors made significant improvements in the manuscript from the original preprint, and the conclusions are now well supported by the evidence presented.

---

## [Author Response]

The following is the authors’ response to the original reviews.

**eLife assessment**
This paper provides useful information about how the ionome of *Arabidopsis thaliana* adapts to very high CO2-levels, backed up by solid evidence and carefully designed studies. However, the broader claims of the paper about climate change and food security - heavily emphasized in the abstract, introduction, and discussion - are inappropriate, as there is no direct link to the presented work.

We sincerely thank you for the work you have done in reviewing our manuscript. We very much appreciate your overall positive assessment of the experimental work as a whole, its value and robustness.

In this revised version, we took on board the majority of your suggestions and your comments. In particular, we understood your critical point about overstating our objectives, which might in turn seem uncorrelated with our results. We fully agree with the comments that have been made on this point. Consequently, we have made substantial modifications and corrections in order to clarify our objectives and their implications: exploring in depth the natural variation of the shoot ionome response to elevated CO2, and generating a valuable resource allowing a better understanding of the genetic and molecular mechanisms involved in the regulation of plant mineral nutrition by the elevation of atmospheric CO2.

We also made modifications in response to the other suggestions, including a clarification of the functional experiments carried out around the function of TIP2;2 in response to elevated CO2. Figure 7 now comprises the comparison between both ambient and elevated CO2 conditions, which is much more informative that what appeared in the previous version.

**Public Reviews:**

**Reviewer #1 (Public Review):**
Summary:The study's abstract, introduction, and conclusions are not supported by the methods and results conducted. In fact, the results presented suggest that Arabidopsis could easily adapt to an extremely high CO2 environment.

We understand the reviewer’s comment. Although our work is considered useful, robust and well designed, we agree with the reviewer's point. We have certainly overemphasized the significance of our work to address the issue of food security in response to rising atmospheric CO2, at the expense of the factual description of the results of our fundamental study of the mechanisms at the interface between CO2 and mineral nutrition. We have clarified this focus by modifying the text of the introduction, objectives and discussion. We hope that these modifications will enable readers to better appreciate the core of this work.

Regarding the last part of the comment, our results do suggest that genetic variation could allow adaptation to rising atmospheric CO2, and our study does indeed aim to identify the extent and basis of this genetic variation.

This study offers good evidence pointing to a genetic basis for *Arabidopsis thaliana's* response to elevated CO2 (eCO2) levels and its subsequent impact on the leaf ionome. The natural variation analyses in the study support the hypothesis that genetic factors, rather than local adaptation, guide the influence of eCO2 on the ionome of rosette leaves in Arabidopsis. However, the manuscript's claim regarding its role in "the development of biofortified crops adapted to a high-CO2 world" (line 23) is overstated, especially given the absence of any analysis on the influence of eCO2 on the seed ionome and Arabidopsis is a poor model for harvest index for any crop. The manuscript, in its current form, necessitates massive revisions, particularly in clarifying its broader implications and in providing more substantial evidence for some of its assertions.

We thank the reviewer for this comment, and we would like to thank the reviewer for the positive appreciation for the identification of genetic basis for *Arabidopsis thaliana's* response to elevated CO2 and its subsequent impact on the leaf ionome. Nevertheless, it is true that the study of the leaf ionome is far from being able to lead to the development of biofortified plants. Some papers described that nutrient harvest index in Arabidopsis is a potential indicator of nutrient use efficiency (for instance, Masclaux-Daubresse and Chardon, Journal of Experimental Botany 2011 or Aranjuelo et al., Journal of Experimental Botany 2013). However, as we did not include any seed ionome data in the paper, we added clear mentions that our analyses were made on leaves (lines 56/57/250/319) and a comment in the discussion section to address this limitation (lines 325-328).

Major Drawbacks and Questions:(1) Evidence for the Central Premise:The foundational premise of the study is the assertion that rising atmospheric CO2 levels result in a decline in plant mineral content. This phenomenon is primarily observed in C3 plants, with C4 plants seemingly less affected. The evidence provided on this topic is scant and, in some instances, contradicts the authors' own references. The potential reduction of certain minerals, especially in grains, can be debated. For instance, reduced nitrogen (N) and phosphorus (P) content in grains might not necessarily be detrimental for human and animal consumption. In fact, it could potentially mitigate issues like nitrogen emissions and phosphorus leaching. Labeling this as a "major threat to food security" (line 30) is exaggerated. While the case for microelements might be more compelling, the introduction fails to articulate this adequately. Furthermore, the introduction lacks any discussion on how eCO2 might influence nutrient allocation to grains, which would be crucial in substantiating the claim that eCO2 poses a threat to food security. A more comprehensive introduction that clearly delineates the adverse effects of eCO2 and its implications for food security would greatly enhance the manuscript.

We partially agree with this comment. The decline in mineral status of C3 plants under conditions of elevated atmospheric CO2 has been widely described in the literature, and specifically documented for the cereal grains. While there are variations in this effect (depending on species, ecotype, cultivar), there is no debate about its acceptance. Here are just a few of the many works describing this effect, both on a global scale and at the level of the individual plant Cotrufo MF (1998) Elevated CO2 reduces the nitrogen concentration of plant tissues. Global Change Biology 4: 43-54; Loladze I (2014) Hidden shift of the ionome of plants exposed to elevated CO(2)depletes minerals at the base of human nutrition. eLife 3: e02245; Myers SS (2014) Increasing CO2 threatens human nutrition. Nature 510: 139-142; Poorter H (1997) The effect of elevated CO2 on the chemical composition and construction costs of leaves of 27 C3 species. Plant, Cell & Environment 20: 472-482 ; Soares JC (2019) Preserving the nutritional quality of crop plants under a changing climate: importance and strategies. Plant and Soil 443: 1-26; Stitt M (1999) The interaction between elevated carbon dioxide and nitrogen nutrition: the physiological and molecular background. Plant, Cell & Environment 22: 583-621; Uddling J (2018) Crop quality under rising atmospheric CO2. Curr Opin Plant Biol 45: 262-267.

In addition to this, the threat to food security posed by this alteration in plant mineral status has also been well described in the literature by several modeling approaches (Beach RH (2019) Combining the effects of increased atmospheric carbon dioxide on protein, iron, and zinc availability and projected climate change on global diets: a modelling study. Lancet Planet Health 3: e307-e317; Ebi KL (2019) Elevated atmospheric CO(2) concentrations and climate change will affect our food's quality and quantity. Lancet Planet Health 3: e283-e284; Medek DE (2017) Estimated Effects of Future Atmospheric CO2 Concentrations on Protein Intake and the Risk of Protein Deficiency by Country and Region. Environ Health Perspect 125: 087002; Smith MR (2018) Impact of anthropogenic CO2 emissions on global human nutrition. Nature Climate Change 8: 834-839; Weyant C (2018) Anticipated burden and mitigation of carbon-dioxide-induced nutritional deficiencies and related diseases: A simulation modeling study. PLoS Med 15: e1002586; Zhu C (2018) Carbon dioxide (CO2) levels this century will alter the protein, micronutrients, and vitamin content of rice grains with potential health consequences for the poorest rice-dependent countries. Sci Adv 4: eaaq1012). To reinforce this point, we have added a sentence and references (lines 30-33). Nevertheless, we understand the reviewer's comment on the nuance to be given to the intensity of this potential threat. We have therefore modified the text, replacing "major threat" by "significant threat" (lines 3 and 29).

We also would like to answer the reviewer’s comment on the potential environmental benefit associated with reduced N and P content in grains (mitigation of N emissions and P leaching). Indeed, if this reduced N and P content results from a lowered use efficiency of soil nutrients by plants, as suggested by several studies (Bloom 2010, Cassan 2023, Gojon 2023 and references therein), this may at the opposite favor N oxides emission and P leaching from the soil.

(2) Exaggerated Concerns:The paper begins with the concern that carbon fertilization will lead to carbon dilution in our foods. While we indeed face numerous genuine threats in the coming decades, this particular issue is manageable. The increase in CO2 alone offers many opportunities for boosting yield. However, the heightened heat and increased evapotranspiration will pose massive challenges in many environments.

While there are indeed multiple threats that we are facing in the coming decades, we don't fully agree with this comment. At present, there's no evidence to say that the negative effect of CO2 on plant mineral content will be manageable. Furthermore, there is compelling evidence that altered mineral nutrition and mineral status of plants will be an important factor limiting the high CO2-induced increase in yield, as will be heat or increased evapotranspiration (see for instance Coskun et al (2016) Nutrient constraints on terrestrial carbon fixation: The role of Nitrogen. J. Plant Physiol. 203: 95-109; Jiang M (2020) Low phosphorus supply constrains plant responses to elevated CO2 : A meta-analysis. Glob Chang Biol 26: 5856-5873 ; Reich PB (2006) Nitrogen limitation constrains sustainability of ecosystem response to CO2. Nature 440: 922-925). Thus, although we do not negate the crucial importance of heat and water stress, we believe it is relevant to study the basic mechanisms responsible for the negative effect of CO2 on plant mineral composition.

Figure 4 in fact suggests that 43% of the REGMAP panel (cluster 3) is already pre-adapted to very high CO2 levels. This suggests annual species could adapt very rapidly.

We agree with the reviewer. However, this suggests that genetic variation exists in some ecotypes to support adaptation to elevated CO2. The purpose of this work is indeed to identify this genetic variation, in order to characterize the mechanisms behind.

(3) Assumptions on CO2 Levels:The assumption of 900ppm seems to be based on a very extreme climate change scenario. Most people believe we will overshoot the 1.5°C scenario, however, it seems plausible that 2.5 to 3°C scenarios are more likely. This would correspond to around 500ppm of CO2. https://www.nature.com/articles/s41597-022-01196-7/tables/4

We agree with the reviewer that the CO2 concentration we used corresponds to a high value in the IPCC projections. That said, this value is currently considered very plausible: the following figure (from Smith and Myers (2018) Nature Climate Change) shows that current CO2 emissions align with the IPCC's most extreme model (RCP 8.5), which would result in a CO2 concentration of around 900 ppm in 2100. Furthermore, nothing allows to exclude the 4°C scenario in the 6th IPCC report.

(4) Focus on Real Challenges:We have numerous real challenges, such as extreme heat and inconsistent rainfall, to address in the context of climate change. However, testing under extreme CO2 conditions and then asserting that carbon dilution will negatively impact nutrition is exaggerated.

While we fully agree that several threats linked to climate change exist, and all deserve to be studied, we find it questionable to consider that the potential effect of high CO2 on the mineral nutrition of plants is not a real challenge. The mineral nutrition of plants is already a current major environmental challenge. This perspective seems to reflect the reviewer's personal opinion rather than an analysis of our work.

In contrast, the FACE experiments are fundamental and are conducted at more realistic eCO2 levels. Understanding the interaction between a 20% increase in CO2 and new precipitation patterns is key for global carbon flux prediction.

Again, we do not fully understand this comment, as the aim of our study was not to perform a global carbon flux prediction, but to unravel genes and mechanisms underlying the negative effect of elevated CO2 on the nutrient content of Arabidopsis rosettes. However, we agree with the reviewer’s comment and with the fact that FACE are useful facilities to explore the CO2 response in more natural environments, and we highlight the fact that the decrease in mineral status of C3 plants has been widely documented in FACE studies. FACE experiments do not facilitate, however, to conduct fully controlled experiments (temperature, rainfall, wind and light intensities are not controllable in FACE), that allow to disentangle the mechanisms by which elevated CO2 regulates the signaling pathways associated with the plant mineral composition. In the longer term, studying the mechanisms we have identified in a more global context of climate change could be highly relevant.

As I look at the literature on commercial greenhouse tomato production, 1000ppm of eCO2 is common, but it also looks like the breeders and growers have already solved for flavor and nutrition under these conditions.

Indeed, tomato is often cultivated in CO2-enriched greenhouses at 1000 ppm. According to the literature, this results in a 20-25% reduction in vitamin C or lycopene, and requires a significantly higher nitrogen and water intake to reach expected sugar levels (Doddrell H (2023) Horticulture Research). In addition, the negative effect of elevated CO2 on tomato nutrient content seems to have significant repercussions on nutrition-health properties (Boufeldja (2023), Molecules).

Conclusion:While the study provides valuable insights into the genetic underpinnings of *Arabidopsis thaliana's* response to elevated CO2 levels, it requires an entirely revised writeup, especially in its abstract, broader claims and implications. The manuscript would benefit from a more thorough introduction, a clearer definition of its scope, and a clear focus on the limits of this study.

We thank the reviewer for the comments made on our manuscript. In addition to the responses that we provide to these comments, we have modified the main text of the introduction, objectives and discussion to take these comments into consideration. We believe that this will significantly improve the manuscript.

**Reviewer #2 (Public Review):**
Strengths:The authors have conducted a large, well-designed experiment to test the response to eCO2. Overall, the experimental design is sound and appropriate for the questions about how a change in CO2 affects the ionome of Arabidopsis. Most of the conclusions in this area are well supported by the data that the authors present.

We thank the reviewer for this positive appreciation.

Weakness:While the authors have done good experiments, it is a big stretch from Arabidopsis grown in an arbitrary concentration of CO2 to relevance to human and animal nutrition in future climates. Arabidopsis is a great model plant, but its leaves are not generally eaten by humans or animals.

We agree with the reviewer’s comment. We recognized that implying a direct contribution of our work to human nutrition in the future climates is overstated, as mentioned by the reviewer 1 as well. This was not an intentional overstatement, as we have always been convinced that our work contributed to the understanding of the basic mechanisms involved in the negative regulation of plant mineral nutrition by high CO2. We have significantly modified the text to correct any misunderstanding of our work’s implication.

The authors don't justify their choice of a CO2 concentration. Given the importance of the parameter for the experiment, the rationale for selecting 900 ppm as elevated CO2 compared to any other concentration should be addressed. And CO2 is just one of the variables that plants will have to contend with in future climates, other variables will also affect elemental concentrations.

We agree with this comment. We added a justification of the high CO2 concentration used in this work in the Material and Methods section (lines 343-344). You can also read the explanation of this choice in the response to the reviewer 1’s point 3.

Given these concerns, I think the emphasis on biofortification for future climates is unwarranted for this study.

Anew, we agree with this comment and we have significantly modified the text to correct any misunderstanding of our work’s implication.

Additionally, I have trouble with these conclusions:-Abstract "Finally, we demonstrate that manipulating the function of one of these genes can mitigate the negative effect of elevated CO2 on the plant mineral composition."-Discussion "Consistent with these results, we show that manipulating TIP2;2 expressions with a knock-out mutant can modulate the Zn loss observed under high CO2."The authors have not included the data to support this conclusion as stated. They have shown that this mutant increases the Zn content of the leaves when compared to WT but have not demonstrated that this response is different than in ambient CO2. This is an important distinction: one way to ameliorate the reduction of nutrients due to eCO2 is to try to identify genes that are involved in the mechanism of eCO2-induced reduction. Another way is to increase the concentration of nutrients so that the eCO2-induced reduction is not as important (i.e. a 10% reduction in Zn due to eCO2 is not as important if you have increased the baseline Zn concentration by 20%). The authors identified tip2 as a target from the GWAS on difference, but their validation experiment only looks at eCO2.

We thank the reviewer for this comment, and we agree with it. It is much more interesting, especially in the context of this paper, to analyze the function of a candidate gene not only in elevated CO2, but in both ambient and elevated CO2. Therefore, we added in Figure 7 data for the expression of TIP2;2 in contrasted haplotypes under ambient CO2, in comparison to those already presented under elevated CO2 (now Fig. 7C and 7D). This showed that TIP2;2 expression is lower in haplotype 0 also under ambient CO2. We also added in Figure 7 (Fig. 7E) the Zn level in WT and tip2;2-1 mutant under ambient CO2, in comparison to those already presented under elevated CO2. This showed that that the tip2;2-1 mutant line did not present any decrease in Zn shoot content in response to elevated CO2, in opposition to what is observed for the WT.

We have added comments associated to these new results in the Results and Discussion sections and in the discussion section (lines 233-242 in the results section, and lines 310-314 in the discussion section).

**Recommendations for the authors:**

**Reviewer #1 (Recommendations For The Authors):**
Reviewer Comments on the Article's Approach to Ionome Analysis(1) Omission of Phosphorus from the Ionome:It's surprising that phosphorus (P) was not measured in the ionome. After nitrogen (N), P is often the most limiting mineral for plant development and yield, making it a significant component of the ionome. Why did the authors omit this crucial element?

We agree with the reviewer that P is an important mineral for plant growth. The absence of data related to P content is due to feasibility constraints rather than oversight. The MP-AES instrument we used to analyze the ionome (except N and C, that we obtained from an Elementar Analyzer) would have required an extra-step and an extra-analysis to obtain data for macronutrient such as P or K. In the context of this large-scale experiment, we faced the necessity to compromise and proceed without these data.

(2) Relationship Between Leaf Ionome and Seed:The manuscript lacks evidence demonstrating the relationship between the leaf ionome and the seed. This connection is vital to establish the study's aims as outlined in lines 20-24. If the central argument is that eCO2 threatens food security, it's essential for the authors to either:Provide evidence that eCO2 induces changes in the ionome profiles of seeds.Show that changes in the rosette leaf ionome lead to alterations in seed ionome profiles.

We agree with the reviewer. Although we know that seed ionome composition of Arabidopsis model accession such as Columbia is indeed negatively affected by eCO2, we do not provide the data that support some of the terms used in lines 20-24. The correspondence between leaf and seed ionome in natural population under eCO2 is certainly a next question that we will address. Therefore, to align our stated objectives with our data, we have modified the sentence in lines 20-24. We also added a comment on this point lines on the discussion section (lines 324-328).

(3) Analysis of Ionome in Rosette Leaves:Why did the authors choose to analyze the ionome specifically in rosette leaves? Is there a known correlation between the ionome profile in rosette leaves and seeds?

See our answer to the above comment.

(4) Experimental Design Comments:The layout of the accession growouts, the methods of randomization, blocking, and controls/checks should be detailed.Were BLUEs (Best Linear Unbiased Estimators) or BLUPs (Best Linear Unbiased Predictors) employed to account for experimental design conditions? If not, it's recommended that they be used.

We thank the reviewer for this comment. A note on replicates has been added in the Method/Plant Material section. Concerning the BLUEs/BLUPs, although I am not familiar with their use, I do not think that these approaches are relevant in our experimental design. Indeed, we pooled 3 to 5 replicates for each accession to measure the ionome (as mentioned in the Method/Ionome analysis section – we realized this was perhaps not clear enough, and thus we reinforced this point in this section). Therefore, we do not have the variance data required to perform BLUEs/BLUPs.

(5) Carbon Dilution Effect:The statement, "The first component of the PCA described a clear antagonistic trend between C content and the change of other mineral elements (Fig. 3B)..." suggests a well-understood carbon dilution effect. These results are anticipated and align with existing knowledge.

We thank the reviewer for this comment. However, this sentence does not relate to the biomass dilution hypothesis referred to by the reviewer. Indeed, the composition of each mineral (C and others) is expressed as a percentage of biomass, not as an absolute value. Therefore, this reflects more a probable effect of the increase in carbon compounds (notably soluble sugars), which could influence mineral composition.

(6) Heritability Estimates:The authors should report both the broad-sense heritability and an estimate of heritability based on a GRM or Kinship matrix.

We thank the reviewer for this suggestion. We are skeptical of using a kinship matrix to estimate heritability in our study. Estimating narrow-sense heritability using a kinship matrix is conceptually based on the infinitesimal model of Fisher, thereby meaning that phenotypic variation is driven by hundreds to thousands of QTLs with small effects. If this is the case, GWAS conducted on several hundred (or even thousands) of genotypes will not be powerful enough to detect such QTLs. Accordingly, estimates of broad-sense heritability based on estimates of variance components can drastically differ from estimates of narrow-sense heritability based on the use of a kinship matrix, as illustrated in the study of Bergelson et al. (2019 Scientific Reports).

(7) Application of the Breeder's Equation:It would be beneficial if the authors applied the breeder's equation to estimate the species' potential rate of response. Based on the allele frequency of the adapted cluster 3 (69 ecotypes or 43% frequency of Figure 3B), it seems plausible that the populations could adapt within 23 generations.

We thank the reviewer for this suggestion. Indeed, it would be really interesting to test whether sub-populations could adapt in comparison with others, and over what period of time. It is nevertheless not possible to do so using the Breeder’s equation in our case, as this requires fitness data under conditions of ambient or elevated CO2 (i.e. production of seeds) to be applied, and we do not have these data at the level of the whole population.

(8) Overall Quality:

In general, the authors have executed a high-quality ionome mapping experiment. However, the abstract, introduction, and discussion should be entirely rewritten and reframed.

We thank the reviewer for the positive evaluation of our experiment. As previously mentioned, we are for the most part in agreement with the comments made about the need to align our stated objectives with our experimental data and conclusions. To do so, we have rewritten part of the abstract, introduction and discussion. The details of these modifications are described in the responses made to each comment.

Here's a line-by-line list of suggestions on writing:Line 30 would read better with a comma after thus (or by replacing thus with therefore and then a comma at the start of the sentence).Line 33 nevertheless would read better in between commas.Lines 45 - 48 sentence is too long, could probably divide it into two.Lines 90 - 94 are hard to interpret, recommend rephrasing for clarity.Line 130 - keep verbs in the past tense for consistency (ran instead of run).Line 194 - what do the authors mean by crossed? I'm inferring they looked at the intersection of DEGs with the list of genes identified by GWA mapping, probably should use a more concise word.There's a concurrent use of the adjective strong (Lines 80, 142, 144, 197, 245). I would advise using a more concise adjective or avoiding its use to let the reader form their own opinion on the data.Lines 174-176 the cited reference (No. 15) is incorrect. The study by Katz et al. (2022) does not provide information on the role of ZIF1 in zinc sequestration mechanisms under elevated CO2 conditions.

We thank the reviewer for these detailed recommendations. We have corrected or rephrased the text according to these suggestions.

**Reviewer #2 (Recommendations For The Authors):**
Technical points:900 ppm as elevated CO2: Given the importance of the parameter for the experiment, the rationale for selection 900 ppm as elevated CO2 compared to any other concentration should be addressed.

We acknowledge the reviewer's point and have previously addressed related aspects earlier in our response. In line with this, we have included a justification for this particular parameter in the Method section.

The authors do not mention what genotype was used for their root/shoot RNAseq experiment.

We thank the reviewer for this comment, and indeed, this information was not mentioned. This is now done, in the Method section.

Line 125: Spelling error "REGMPA".

This has been corrected.

Line 338: Removal of outlier observations - "Prior to GWAS and multivariate analyses such as PCA or clustering, mineral composition measures were pre-processed to remove technical outliers". The authors should mention the exact number of outliers that were removed and what the explicit criteria were for removal.

The number of outliers removed from each dataset is now indicated in Supplemental Table 7 (this is cited in the Method section). The explicit criteria used for this analysis is actually mentioned in the corresponding Method section: “the values positioned more than 5 median absolute deviations away from the median were removed from the dataset”.

Line 379: "Lowly expressed genes with an average value across conditions under 25 reads were excluded from the analysis". Providing information about the number of the lowly expressed genes that were removed from the analysis can help with the interpretation of the likelihood of the candidates selected being correct.

This is a standard procedure in RNAseq analysis. It avoids many false positives in the differential analysis of gene expression based on ratios (where a very small number in the denominator can lead to a very high variation in expression, of no real significance). For information, this step led to the removal of 11607 and 10121 genes for the shoot and root datasets.

Line 384: It's not clear how many biological replicates were used.

This has been corrected.

Additional comment: We have also become aware of a confusion concerning one of the candidate genes located close to GWA peaks: line 180 of the first version, we mentioned CAX1 (AT1G16380) for its role on nutrient deficiency response. There are actually two genes annotated as CAX1 in TAIR (both are cation exchangers), but the one involved in nutrient deficiency response is AT2G38170. We therefore removed the sentence mentioning AT1G16380/CAX1 as a potential candidate gene.